# Signaling Role of Pericytes in Vascular Health and Tissue Homeostasis

**DOI:** 10.3390/ijms25126592

**Published:** 2024-06-15

**Authors:** Antonietta Fazio, Irene Neri, Foteini-Dionysia Koufi, Maria Vittoria Marvi, Andrea Galvani, Camilla Evangelisti, James A. McCubrey, Lucio Cocco, Lucia Manzoli, Stefano Ratti

**Affiliations:** 1Department of Biomedical and Neuromotor Sciences, University of Bologna, Via Irnerio 48, 40126 Bologna, Italy; antonietta.fazio2@unibo.it (A.F.); irene.neri3@unibo.it (I.N.); foteinidionysi.koufi@unibo.it (F.-D.K.); mariavittoria.marvi2@unibo.it (M.V.M.); andrea.galvani6@unibo.it (A.G.); camilla.evangelisti@unibo.it (C.E.); lucio.cocco@unibo.it (L.C.); lucia.manzoli@unibo.it (L.M.); 2Department of Biomolecular Sciences, University of Urbino “Carlo Bo”, 61029 Urbino, Italy; 3Department of Microbiology and Immunology, Brody School of Medicine, East Carolina University, Greenville, NC 27834, USA; mccubreyj@ecu.edu

**Keywords:** pericytes, heterogeneity, signaling pathways, angiogenesis, tissue regeneration, tissue homeostasis

## Abstract

Pericytes are multipotent cells embedded within the vascular system, primarily surrounding capillaries and microvessels where they closely interact with endothelial cells. These cells are known for their intriguing properties due to their heterogeneity in tissue distribution, origin, and multifunctional capabilities. Specifically, pericytes are essential in regulating blood flow, promoting angiogenesis, and supporting tissue homeostasis and regeneration. These multifaceted roles draw on pericytes’ remarkable ability to respond to biochemical cues, interact with neighboring cells, and adapt to changing environmental conditions. This review aims to summarize existing knowledge on pericytes, emphasizing their versatility and involvement in vascular integrity and tissue health. In particular, a comprehensive view of the major signaling pathways, such as PDGFβ/ PDGFRβ, TGF-β, FOXO and VEGF, along with their downstream targets, which coordinate the behavior of pericytes in preserving vascular integrity and promoting tissue regeneration, will be discussed. In this light, a deeper understanding of the complex signaling networks defining the phenotype of pericytes in healthy tissues is crucial for the development of targeted therapies in vascular and degenerative diseases.

## 1. Introduction

Pericytes are a heterogeneous cellular population of the vascular system, crucial for regulating cellular mechanisms involved in the maintenance of blood vessel stability and functionality [1]. These cells are situated within microvascular niches, playing a crucial role in preserving tissue homeostasis, particularly in specific areas such as the brain, retina, adipose tissue, myocardium and skeletal muscle. This balance is guaranteed by intricate interactions of pericytes with other cells of the microvascular environment, forming a complex network of support. Consequently, disruptions of these interactions can lead to vascular dysfunctions, thereby contributing to the onset and progression of various diseases, underscoring the critical role of pericytes in vascular health [2].

Moreover, pericytes display significant potential in regenerative medicine due to their stem cell-like abilities, which enable them to differentiate into multiple cell types and significantly contribute to tissue repair and regeneration [3,4,5]. With the increasing recognition of disorders associated with pericyte dysfunction, these multitasking cells are emerging as promising candidates for developing targeted therapeutic approaches in both vascular and non-vascular contexts [6].

However, defining pericytes as stem cells presents challenges due to limited scope of supporting studies, which primarily rely on in vitro experiments or observations from animal models where previously cultured pericytes have been transplanted [7]. Indeed, these artificial conditions may stimulate progenitor potential that may not otherwise manifest in vivo. In line with these findings, recent research by Guimarães-Camboa et al. [8] demonstrated, by using a Tbx18-CreERT2 transgenic mouse model, that pericytes do not transdifferentiate into different cellular populations during aging or after injury in organs like the brain, heart, and skeletal muscle. Hence, this finding confirms that the potential of pericytes may vary depending on the context, underscoring the significance of expanding our understanding of pericyte biology within their natural environment.

The functional regulation of pericytes is mediated by a complex network of signaling pathways. These signaling cascades control pericyte activity and orchestrate their interactions with neighboring cells, such as endothelial cells and other vascular components, which are vital for preserving tissue equilibrium. Key pathways, including platelet-derived growth factor beta (PDGFβ)/ platelet-derived growth factor receptor beta (PDGFRβ), transforming growth factor beta (TGF-β), and Notch, are essential for the recruitment, adhesion, and functional maturity of pericytes [9]. Additionally, the Akt/mTOR, Forkhead box O (FOXO), vascular endothelial growth factor (VEGF), and Smad pathways play significant roles in modulating pericyte responses to environmental stimuli and stress, influencing their capacity to transition to different cell types under both physiological conditions, such as angiogenesis, and pathological states, such as tissue injury [10]. This intricate and interconnected interplay among molecular pathways establishes a dynamic framework that allows pericytes to respond and integrate signals from multiple sources, thereby adapting their behavior to support vascular stability and facilitate tissue repair.

This review aims to provide a comprehensive overview of the signaling pathways that influence pericyte functions within the physiological system. This review will begin with an introduction to the fundamental characteristics of pericytes, followed by an exploration of their specific roles and the complex signaling networks that regulate their activities. Therefore, exploring the multifaceted nature of pericyte could pave the way for targeted interventions in a spectrum of pathological conditions, thereby ameliorating vascular dysfunctions and promoting regenerative healing processes.

## 2. Identity and Heterogeneity of Pericytes

Pericytes are multifunctional cells strategically located within the vascular niche, primarily recognized for their pivotal role in maintaining vascular stability and regulating blood flow. Originally identified by Charles-Marie Benjamin Rouget in 1873 and referred to as ‘Rouget cells’, these cells were later named ‘pericytes’ in 1923 by Zimmerman to highlight their enveloping position around capillaries and post-capillary venules [11]. Based on their morphology and location, pericytes are divided into three subtypes: ensheathing pericytes on pre-capillaries, thin-stranded pericytes on capillaries, and stellate pericytes on post-capillaries [12]. They differ in their primary and secondary processes and vessel coverage, but all subtypes interact with endothelial cell junctions to regulate blood vessel functionality and stability. Notably, the ratio of pericytes to endothelial cells varies significantly, ranging from 1:100 in skeletal muscle to 1:3 in the central nervous system, and 1:1 in the retina [13].

Pericytes have different embryonic origins across tissues, reflecting their diverse roles from vascular development to tissue repair. Lineage tracing studies showed that pericytes in the cephalic region and thymus are predominantly ectomesenchymal [14,15,16], while in the lungs, heart, liver, and gut, pericytes mainly derive from mesothelium [17,18,19,20].

Phenotypically, pericytes are identified through a combination of cell surface and intracellular markers, which vary by their tissue location and developmental source. The most recognized markers include PDGFR-β [21,22], a receptor essential for pericyte survival and proliferation [23], and Neural/Glial Antigen 2 (NG2), a surface proteoglycan expressed throughout all stages of pericyte development [24,25,26]. Studies of PDGFR-β and NG2 knockout in transgenic mice have shown that the absence of these markers leads to a reduced pericyte coverage and significant deficits in vascular integrity, emphasizing their critical role in maintaining vessel structure [27,28].

Pericytes are identified using techniques such as immunohistochemistry, flow cytometry, and microscopy, which are essential for verifying their phenotype. Despite considerable advances in their characterization, the heterogeneity of pericytes across various tissues and organs presents ongoing challenges, as these markers are not unique to pericytes but are also expressed by other cell types [29,30]. Consequently, distinguishing pericytes from other mural cells in various vascular niches necessitates the combination of additional markers (Table 1). A recent study using single-cell RNA sequencing analysis on a mouse model has recognized several tissue-specific pericyte markers. The potential markers identified include Kcnk3 in the lung, Rgs4 in the heart, Myh11 and Kcna5 in the kidney, and Pcp4l1 in the bladder, with Higd1b being common to both lung and heart. These markers were further validated by comparing data from the Human Lung Cell Atlas and human heart single-cell RNAseq databases, revealing an overlapped expression of these markers in human tissues [31]. Moreover, based on a recent research protocol [32], other markers are used to distinguish pericytes by immunohistochemistry, such as CD146 [33], CD34 [34] and αSMA [35]. However, these markers are also expressed by endothelial cells, hematopoietic stem cells, vascular smooth muscle cells and myofibroblasts, respectively. It means that these targets lack high specificity only for pericytes and in all tissues, but can be used to exclude the presence of other cell types. Thus, this approach helps to effectively characterize pericytes by narrowing down the cell population under examination. In response to these complexities, current research is intensely focusing on refining these targets to develop more specific, cost-effective, and efficient methods.

## 3. Physiological Functions of Pericytes

Pericytes are essential mural cells that significantly contribute to maintaining the blood vessels integrity and orchestrating tissue regeneration and repair processes. They intricately regulate blood flow, actively participate in angiogenesis, and leverage their potential stem cell-like properties to facilitate tissue recovery and regeneration. These multifaceted roles draw on pericytes’ remarkable ability to respond to biochemical cues, engage in interactions with neighboring cells, and adapt to changing environmental conditions. This section of the review explores the diverse and indispensable contributions of pericytes in various physiological contexts.

### 3.1. Blood Flow Regulation

Pericytes are involved in the regulation of microvascular blood flow by dynamically respond to a range of biochemical stimuli, including neurotransmitters and hormones, as demonstrated in retina and cerebellar samples [36]. The regulation of blood flow is intricately ensured through the interactions of pericytes with surrounding cellular components. Specifically, in the extensively studied neurovascular unit, the interaction between pericytes, endothelial cells, microglia, neurons and astrocytes is essential to maintain vascular stability and regulate blood flow [37]. This cellular network is significant in the brain, where pericytes contribute to neurovascular coupling and the integrity of the blood–brain barrier (BBB). Indeed, they regulate the exchange of substances and maintain cerebral blood flow, which is crucial for protecting neural tissue [38]. Notably, neuronal activity and the neurotransmitter glutamate induce the release of factors triggering the relaxation of pericytes. Conversely, pathologic conditions, such as ischemia, have been associated to pericyte constriction and death, leading to capillary constriction, BBB damage, and decreased blood flow [39]. Consequently, avoiding the constriction and death of pericytes may help mitigate the decrease in blood flow, which in turn could protect neurons from damage after a stroke. Hence, their dysfunction is linked to a range of microcirculatory disorders such as stroke, Alzheimer’s disease, and diabetic retinopathy, where compromised capillary flow contributes to tissue hypoxia and accelerates the progression of these conditions [40].

Interestingly, Hill et al. [41] have demonstrated in an in vivo mouse model that the cerebral capillary contractility is mediated by arteriolar smooth muscle cells rather than pericytes. Indeed, it is supposed that pericytes mostly take part in the vascular system’s signal transmission and coordination within the vascular compartment. Instead, Hartmann et al. [38] have shown that pericytes play a role in regulating basal capillary flow resistance by contraction within the brain. These contradictory results across studies may stem from variations in the blood vessel and mural cell types examined.

Hence, resolving these divergent observations is crucial for elucidating the precise roles of pericytes in cerebral blood flow regulation, especially in light of findings indicating arteriolar smooth muscle cells as major regulators of the contraction mechanism. Understanding the distinct contributions of each cell type involved is essential for developing targeted therapeutic interventions aimed at modulating cerebral perfusion dynamics. This comprehension holds particular significance in the context of neurological conditions like stroke.

### 3.2. Angiogenesis

Angiogenesis is a physiological and pathological process of new blood vessel formation from pre-existing ones. Central to this process are pericytes, strategically positioned along the vascular wall, where they closely interact with endothelial cells [42]. These interactions are crucial for the formation and stabilization of new vascular networks that can vary depending on the specific microenvironment and signaling cues present in different tissues or pathological conditions, such as tumor environment.

Pericytes not only support but actively participate in the angiogenic process through dynamic changes in their phenotype. They migrate towards areas of angiogenic activity, line up with the developing vessels and establish direct contact with endothelial cells. This coordinated behavior is essential for the structural integrity and functionality of newly formed capillaries [43]. Evidence of their pivotal role is highlighted in various in vitro models related to the BBB, where pericytes co-cultured with endothelial cells and astrocytes effectively reorganize themselves by contributing to the formation of stable capillary-like structures [44].

Further emphasizing their role, pericytes are among the first cells to populate newly vascularized areas [45]. Notably, hypoxic conditions stimulate their migration and trigger angiogenic transformation by reducing their processes and enlarging their somatic volume [46,47]. These adaptations are crucial for orchestrating the expansion of newly formed blood vessels, enhancing endothelial cell survival, and promoting the formation of tubular structures [48,49].

Moreover, the aforementioned process is critical not only in the brain but extends to other tissues such as skeletal muscle, where angiogenesis manifests in two forms: sprouting and nonsprouting [50]. In sprouting angiogenesis, pericytes actively proliferate and migrate, guiding endothelial cells to form new vascular branches. This process involves significant structural remodeling, including capillary basement membrane degradation by matrix metalloproteinases (MMPs). Conversely, in nonsprouting angiogenesis, pericytes maintain stability, aiding the division of existing vessels without breaking down the basement membrane, but synthesizing new basement membrane components, helping the formation of new channels through a process called intussusception. These contrasting roles highlight pericytes’ adaptability in different angiogenic contexts, either actively shaping new vascular pathways or conserving structural integrity during vessel splitting [50].

In conclusion, their ability to guide and sustain the growth of new blood vessels highlights their significance in vascular health and pathology, making them key targets for therapeutic strategies aimed at modulating angiogenesis in vascular complications.

### 3.3. Stem Cell Potential and Tissue Regeneration

Pericytes, recognized as mesenchymal stem cells (MSCs) within microvascular niches of various tissues, demonstrate significant stem cell potential, positioning them as central players in regenerative medicine. In the central nervous system, pericytes share functional characteristics with MSCs, including the expression of specific stem cell markers like CD44, CD73, CD90, and CD105. Indeed, they possess the capacity to differentiate into neuron-like cells, astrocytes, and oligodendrocytes [51], highlighting their potential in developing regenerative strategies for neurological disorders [52,53].

Similarly, in the skeletal muscle, studies suggest that pericytes may differentiate into myogenic lineages, crucial for muscle regeneration (Figure 1) [48,54]. These cells are dominantly quiescent within their vascular niche but can be activated by muscle injury or other pathologic conditions leading to proliferation and myogenic differentiation [55].

Recent research using transgenic mice has further refined our understanding of pericyte diversity, categorizing them into two distinct types based on marker expression and functional roles. Type-1 pericytes (Nestin−/NG2+) primarily differentiate into adipocytes and contribute to fat accumulation, whereas Type-2 pericytes (Nestin+/NG2+) transform into muscle cells, significantly aiding in muscle regeneration [55,56,57]. Beyond that, pericytes also exhibit osteogenic capabilities, important for bone healing and regeneration, by showing considerable contributions to bone repair, as demonstrated in animal models of muscle pockets spine fusion, calvaria and nonunion fracture [58]. These capabilities further extend to the treatment of skeletal defects, leveraging their ability to modulate the osteoprogenitor environment through paracrine signaling [59]. Additionally, it is reported that pericytes may differentiate into other cell lineages, such as macrophages and adipocytes, influencing processes like immune regulation and adipogenesis respectively [60,61].

However, the multipotency of pericytes in vivo remains unclear and a subject of ongoing investigation, due the existence of controversial and conflicting data. Studies have shown that in the heart, brain, skeletal muscle, and adipose tissue, pericytes do not appear to contribute to the generation of other cell lineages during aging and in pathological conditions [8], as instead evidenced in the above research. Furthermore, research has shown that differentiation into another lineage could be highly cell specific. For instance, stromal cells composing spinal cord scar tissue originate from a specific pericyte subtype [62]. Therefore, it is important to emphasize that these cells’ plasticity may be influenced by in vitro circumstances by forming an artificial culture. However, these findings should not limit the exploration of pericytes for regenerative purposes, but rather underscore the need for further comprehensive investigations. Collectively, the widespread presence and multifunctional nature of pericytes across various tissues offer promising avenues for the development of cell-based therapies. Their unique position within the vascular niche, combined with their properties and ability to interact with a multitude of cell types, underscores their potential in future research and clinical applications aimed at exploiting their therapeutic capabilities [63,64].

## 4. Signaling Pathways Coordinating Pericyte Functions

Pericytes are multifunctional cells governed by intricate networks of signaling pathways (Figure 2). This complex interplay of interconnected axes influences each other and activates downstream molecules involved in a range of physiological and pathological conditions. Hence, a deeper understanding of these dynamic interactions offers promising pathways for targeted therapies in vascular or other diseases, underscoring their significant potential for clinical advancements. This section of the review explores the transduction signaling governing pericyte functions in physiological contexts.

### 4.1. Signaling of Blood Flow Regulation

Pericytes are pivotal for maintaining vascular stability, interacting with endothelial cells through crucial signaling pathways that preserve the integrity and functionality of blood vessels. Central to this cell interaction is the PDGFβ/PDGFRβ signaling pathway, primarily binding pericytes to capillaries and ensuring vascular stability by reducing permeability [9,65]. Disruptions in PDGFβ/PDGFRβ axis are responsible for loss of pericytes, notably affecting organs like the brain, kidneys, and lungs, while liver pericytes remain unaffected, as demonstrated in genetically modified in vivo models [27]. Recent studies have also emphasized the importance of PDGF-β and hypoxia-inducible factor-1alpha (HIF-1α) for the crosstalk between brain microvascular endothelial cells and brain vascular pericytes in maintaining BBB integrity under ischemic conditions [66]. This signaling crosstalk is crucial for protecting against ischemic damage. Similarly, HIF-2α has played a significant role in pulmonary vascular remodeling in conditions such as pulmonary arterial hypertension (PAH), affecting pericyte–endothelial cell interactions and enhancing vessel contractility. Indeed, the overexpression of HIF-2α has been associated with a greater contractility and impaired endothelial cells-pericytes interaction [67].

Of note, TGF-β signaling is involved in recruiting pericytes to endothelial sites, which aids in vessel maturation and stability [68]. Among the TGF-β type I receptors, only activin receptor-like kinase (Alk)-5 promotes vessel maturation by inducing the phosphorylation of Smad2/3 and triggering differentiation into smooth muscle cells [69]. Further enriching this network, Integrin αvβ8 activates latent TGF-β in human astrocytes or freshly dissociated fetal brain cells, influencing endothelial functions and stabilizing cerebral vessels. This integrin is pivotal in regulating genes like plasminogen activator inhibitor-1 and thrombospondin-1, which are essential for vessel differentiation and stability, thereby serving as a central regulator of cerebral vascular homeostasis [70].

Additionally, the complex interplay between TGFβ, Smad, Notch, and sphingosine-1-phosphate (S1P) pathways regulates crucial adhesion molecules like N-cadherin, which mediates interactions between endothelial cells and pericytes [71,72]. Indeed, changes in these pathways, such as the deletion of Smad4, result in reduced pericyte interactions and the downregulation of Notch receptors and N-cadherin in cerebrovascular endothelial cells [73]. Moreover, the interaction of bioactive S1P with its receptor affects N-cadherin’s role in cell recruitment, further inducing vascular dysfunction and increased permeability [74].

These insights demonstrate the intertwined nature of the mentioned signaling pathways in vascular maturation and integrity. Understanding the complex interactions not only underscores the intricate nature of vascular biology but also broadens the knowledge on targeted therapeutic interventions in vascular diseases, offering potential for significant clinical advancements.

### 4.2. Signaling of Angiogenesis

Angiogenesis is a dynamic process orchestrated by a complex interplay of signaling pathways, growth factors, extracellular matrix proteins, and adhesion molecules. These elements coordinate cellular proliferation and migration necessary for capillary sprouts formation [48]. Studies have reported that pericytes actively participate in angiogenesis through the action of molecules such as PDGF-β, TGF-β, VEGF, Angiopoietin 1 (Ang-1), and S1P. PDGF-β, secreted by endothelial cells, specifically binds PDGFRβ receptors on pericytes, promoting their migration toward developing vessels. However, studies on PDGF-β- and PDGFR-β-deficient mouse models have revealed significant vascular anomalies, including lack of pericytes, endothelial hyperplasia and increased brain vascular permeability, as evidenced by abnormal distribution of junctional proteins and increased transendothelial permeability [75].

Furthermore, the regulation of angiogenesis involves complex interactions between pericytes and endothelial cells, also facilitated through the Notch3 and Jagged-1 signaling pathways. This axis is crucial for strengthening cell–cell contact by inhibiting endothelial cell migration and proliferation while also modulating PDGF-β expression [76]. The complexity of this interaction is further exemplified by the Akt1-Notch3/YAP-Ang-1/2 cascade in vascular smooth muscle cells, which significantly impacts vascular stability and function [10]. Studies involving Akt1-deficient mice have demonstrated that the absence of Akt1 results in delayed retinal angiogenesis, characterized by defective endothelial cell proliferation, impaired pericyte recruitment, and reduced coverage of the endothelium by vascular smooth muscle cells. Indeed, silencing Akt1 specifically in vascular smooth muscle cells results in reduced Notch3 activation, which disrupts the balance between Ang-1 and Ang-2. This disruption prevents endothelial sprouting and exacerbating vascular complications, particularly under diabetic conditions [10]. It is important to note that the balance between Ang-1 and its antagonist Ang-2 is essential for maintaining vascular homeostasis, with Ang-2 promoting instability when overriding Ang-1 signaling [77]. Indeed, disruption in Ang-1 expression due to genetic modifications on the chicken ovalbumin upstream promoter-transcription factor II (COUP-TFII) leads to compromised angiogenesis in tumors and vascular defects [78].

Pericytes produce TGF-β, another crucial regulator that exerts a dual role in angiogenesis: it inhibits endothelial cell proliferation to control excessive sprouting while simultaneously stimulating pericyte differentiation, thus supporting the structural integrity of newly formed vessels [79]. The significance of TGF-β extends to CNS angiogenesis, where the knockdown of the G protein-coupled receptor 124 (Gpr124) gene, a member of the long N-terminal group B family of G protein-coupled receptors, results in defective vascular development characterized by delayed vascular penetration and hemorrhage [80].

Growth factors like basic fibroblast growth factor (bFGF) and vascular endothelial growth factor (VEGF) are essential for pericyte migration and proliferation. Additionally, they play a crucial role in pericyte recruitment and proper adhesion to endothelial cells, influencing angiogenesis and overall vessel stability [45]. In the postnatal retinal vasculature, pericytes are shown to regulate this axis by expressing vascular endothelial growth factor receptor 1 (VEGFR1), which enhances endothelial sprouting. This modulation of VEGF activity is critical, as both genetic depletion of pericytes and targeted disruption of VEGFR1 in pericytes lead to angiogenic defects similar to those observed after intraocular injection of VEGF-A. These findings highlight the essential roles of pericytes in vascular formation and integrity, which are often compromised in diseases like diabetic retinopathy [81]. Moreover, under hypoxic conditions, such as those occurring in ischemic tissues, pericytes become even more active. They release VEGF-A, enhancing the survival and migratory capacity of endothelial cells, and thereby facilitating vascular sprouting [82]. This release is a critical response to oxygen deficiency, driving the angiogenic process to restore adequate blood flow. The activation of the PI3K/AKT/mTOR pathway is significant in these processes, facilitating VEGF secretion and representing a potential therapeutic target for inducing angiogenesis, in particular post-stroke [83]. During vascular remodeling, pericytes present a notable decrease in PI3K signaling represented by an early pericyte maturation, as demonstrated by genetic PI3Kβ inactivation in mouse models [84]. Conversely, the release of PI3K signaling by means of PTEN deletion delayed pericyte maturation, suggesting its crucial role in vessel remodeling during angiogenesis. Targeting key molecules in this pathway can manipulate angiogenic responses, as demonstrated by the effects of the traditional Chinese compound named Astragaloside IV, which promotes angiogenesis by activating the PI3K/Akt signaling pathway in conditions such as myocardial infarction [85]. It has been also demonstrated to provide neuroprotection in ischemic stroke through mechanisms that reduce oxidative stress, inflammation, and apoptosis, further validating its pro-angiogenic capabilities in in vitro studies [86].

Therefore, pericytes are not just passive structural elements but are active participants in the angiogenic process, crucial for both the formation and stabilization of new blood vessels. Their various interactions with ECs through multiple signaling pathways underscore their vital role in vascular biology, highlighting their potential as therapeutic targets in diseases characterized by abnormal angiogenesis and vascular instability.

### 4.3. Signaling of Stem Cell Potential and Tissue Regeneration

Pericytes play a crucial role in regenerative mechanisms and differentiation through complex cellular signaling pathways that influence their multipotency and fate decisions. They maintain their stemness by interacting with the basement membrane component laminin, which supports their identity during angiogenesis and suppresses premature differentiation, while also responding to various environmental cues and metabolic changes [87]. For instance, hypoxia enhances their glycolytic activity, helping to preserve multipotency by inhibiting differentiation into specific lineages. This capacity for multipotency is modulated by signaling pathways and transcription factors like FOXO, which regulates metabolic states and antioxidant defenses essential for self-renewal and quiescence of stem cells. Of note, the depletion of FOXO3A leads to diminished self-renewal capacities and increased differentiation in various cellular lineages, including neural, hematopoietic and satellite cells [88,89].

Moreover, antioxidant treatments have been shown to enhance self-renewal capacities of FOXO3-deficient stem cells, highlighting the significant role of oxidative stress regulation in stem cell fate decisions [89,90]. In pericytes, FOXO3A also promotes quiescence, partly by upregulating Notch1 and Notch3 signaling [91], which in turn influences pericyte migration and capillary growth through mechanisms such as the Ang-2-induced reduction in Tie-2 activation [92]. All these findings suggest that FOXO3A maintains quiescence and modulates the differentiation pathways of pericytes by responding to oxidative stress, a key determinant of pericyte fate during regeneration.

Pericytes also preserve their quiescent state through the cell adhesion protein N-cadherin, which facilitates heterotypic interactions between endothelial cells and pericytes [71]. Notably, N-cadherin contributes to sequester β-catenin at the plasma membrane, potentially attenuating proliferation and impacting c-Myc expression [93].

In addition, it has been demonstrated that the heightened expression of the growth factor TGF-β2 prompted pericytes to migrate into the subretinal space and undergo a transition into myofibroblasts through the Smad2/3 and Akt/mTOR pathways [94]. This is a key event that significantly contributes to the progression and exacerbation of subretinal fibrosis in neovascular age-related macular degeneration.

Hence, pericytes intricately are part of a dynamic interplay, balancing between retaining their stemness and transitioning towards specific differentiation pathways, influenced by microenvironment, metabolic dynamics, and signaling cascades. Together, these elements assemble the diverse roles of pericytes in physiological processes and underline their promising applications in regenerative medicine.

## 5. Conclusions

Pericytes are a unique cellular population within the vascular system, garnering significant interest in scientific and medical research for their critical roles in maintaining vascular stability, facilitating tissue regeneration, and their involvement in various disease processes. A crucial aspect of pericyte research involves understanding the complex signaling pathways that govern their functions and behaviors, particularly in vascular stability, angiogenesis, and their differentiative ability. Hence, exploring signaling cascades such as PDGFβ/PDGFRβ, TGF-β, and Notch and others could provide insights into the molecular mechanisms underlying both physiological and pathological conditions, offering potential targets for therapeutic interventions.

The ability of pericytes to enhance vascular stability and prevent leakage makes them essential in tissue regeneration and engineering. Leveraging their potential stem cell-like properties, researchers are investigating pericytes in the development of bioengineered tissues for transplantation, particularly in treating degenerative diseases where tissue repair and regeneration are crucial. However, further studies will be needed to fully support this potential capability. Furthermore, pericytes play a crucial role in maintaining the integrity of vascular barriers such as the BBB. This aspect is increasingly important as drug delivery systems evolve, presenting new opportunities to develop targeted delivery mechanisms capable of efficiently crossing these barriers, especially for the treatment of neurological conditions.

In summary, the multifunctional nature of pericytes and their involvement in intricate signaling networks position them as focal points for ongoing and future research. Their central role in both physiological processes and potential therapeutic applications makes them critical for the understanding and treatment of a wide range of medical conditions.

## Figures and Tables

**Figure 1 ijms-25-06592-f001:**
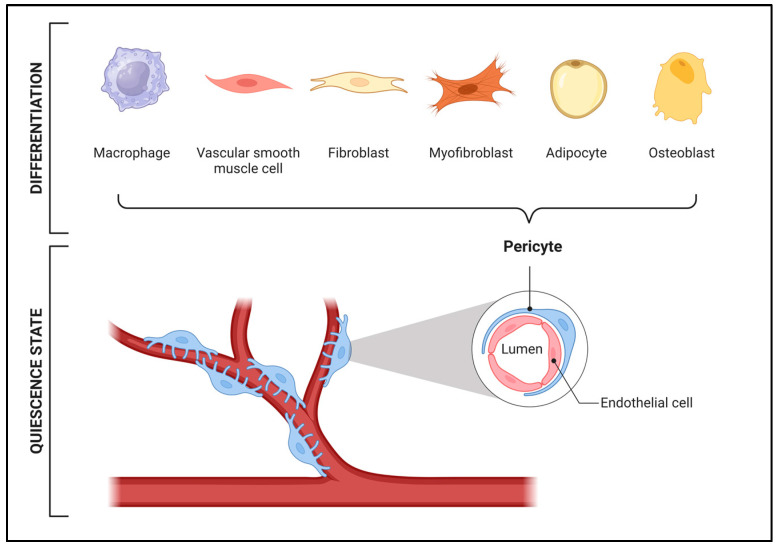
Pericyte differentiation potential in skeletal muscle regeneration. The maintenance of vessel stability and repair is guaranteed by the intricate organization of pericytes interacting with endothelial cells. Moreover, pericytes possess the capability to differentiate into diverse cellular populations, offering promising avenues for therapeutic interventions aimed at enhancing muscle tissue regeneration and repair.

**Figure 2 ijms-25-06592-f002:**
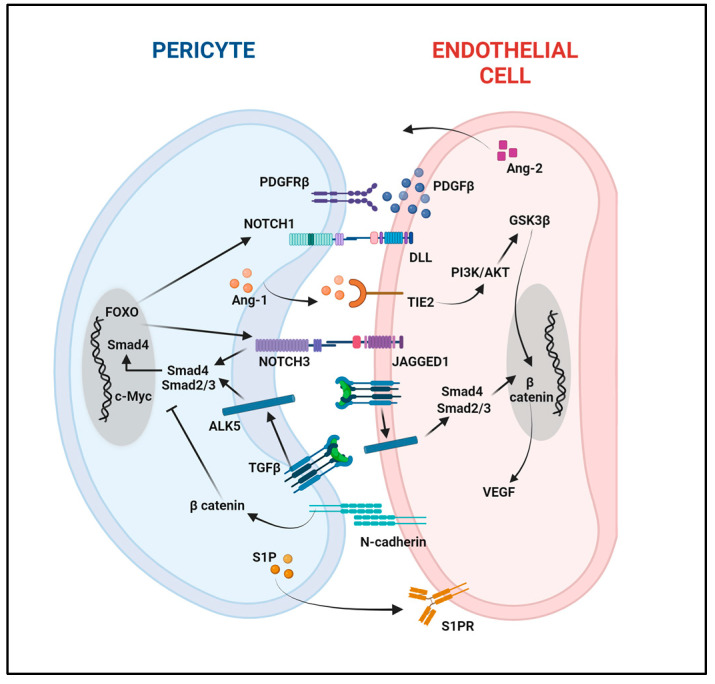
Signaling pathways coordinating pericyte–endothelial cell interactions. ALK5 (activin receptor-like kinase 5); Ang-1 (Angiopoietin 1); Ang-2 (Angiopoietin 2); DLL (Delta-like 1); FOXO (Forkhead box O); GSK3β (Glycogen synthase kinase-3 beta); PDGFβ (Platelet-Derived Growth Factor Beta); PDGFRβ (Platelet-Derived Growth Factor Receptor Beta); S1P (Sphingosine-1-phosphate); S1PR (Sphingosine-1-phosphate Receptor); TGFβ (Transforming growth factor beta); TIE2 (Angiopoietin-1 receptor); VEGF (Vascular Endothelial Growth Factor).

**Table 1 ijms-25-06592-t001:** Markers commonly used for identifying pericytes in different tissues.

	Role	Expression	Ref.
**PDGFR-β**	Cell surface tyrosine kinase receptor involved in stability and integrity of blood vessels.	Localized in pericytes on the small blood vessels, particularly in capillaries and venules.	[21,22]
**NG2**	Type I transmembrane protein involved in vascular remodeling and stability, cellular migration and proliferation.	Localized on oligodendrocyte precursor cells (OPCs), as well as pericytes associated with arterioles and capillaries.	[24,25,26]
**CD146**	Cell adhesion molecule involved in cell–cell interaction and vascular remodeling.	Expressed in pericytes of skeletal muscle, brain, pancreas, adipose tissue, and placenta. Additionally, it is also expressed in endothelial cells.	[33]
**CD34**	Cell adhesion protein implicated in vascular stabilization, cell adhesion and migration.	Expressed in pericytes, as well as in hematopoietic stem cells, endothelial cells and fibroblasts.	[34]
**α-SMA**	Contractile protein involved in cell contraction, structure and integrity of the microcirculation.	Located in pericytes on pre-capillary arterioles and post-capillary venules. Additionally, it is also expressed in vascular smooth muscle cells and myofibroblasts.	[35]

## Data Availability

No new data were created or analyzed in this study.

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
