# Peer review of "Signaling Role of Pericytes in Vascular Health and Tissue Homeostasis"

_ijms, 2024, doi:10.3390/ijms25126592_

Round 1
Reviewer 1 Report
Comments and Suggestions for Authors
In the review manuscript submitted by Fazio et. al, the authors present an overview of pericyte (PC) identity (including markers), PC functional roles (including contractility, angiogenesis, and stemness), and signaling pathways (related to blood flow, angiogenesis, and stemness). A thorough review each of these areas of PC biology would be a significant contribution to the field in order to provide clarity to a landscape full of debate and controversy. However, the current review does not, in this reviewers perspective, thoroughly review the topics of focus. Over 1/3 of the references provided were of other reviews. This is unacceptable and reflects a lack of effort and sincerity. While signaling mechanisms were better cited and more thoughtfully approached, some sections of the review presented one-side only of controversial topics, further contributing to confusion in the field. In preparing a review, it is necessary to present a thorough overview of the current state of the field, supported by original research publications. The impact of the current review is lacking due to inadequate and limited literature review and citations. For example:
1. Wherever possible, it is critical to cite original research papers rather than reviews.
2. Lines 41-42 – stemmness of pericytes is controversial, and is not widely accepted among many pericyte scientists. It is important to present both sides of this topic. For instance, PC stemness has been demonstrated as an artifact of cell-culture (Nuno Guimaraes-Camboa et al., 2017, Elena Cano et al 2017) – an environment in which many cell types drift, despite being fully differentiated. Cited with a review.
3. Lines 76 – The authors describe PC morphology as an “elongated cell body aligned with the direction of blood flow”, cited with a review. However, three recognized morphological subtypes of PCs are described in the literature, based on vessel location.
4. Lines 82-84 are almost word for word, including references, taken from a 2021 review by Girolamo et al. (although the authors of the review under submission state (incorrectly) neuroectodermal rather than ectomesenchyme. PCs were reported to be derived via neuroectoderm in a 2002 paper by Korn and Kurz, which when examined, actually describes SMCs, not PCs.
5. Line 88-89 – NG2 is not just an identifier of developmental PC. It is a prominent PCs marker throughout all stages.
6. Lines 85-98, and marker Table. The authors include markers that are not widely used or accepted for PC identification, and thus detract from the value of this section. While NG2 and PDGFRβ are prominent PC markers, PC alpha-SMA expression is variable, and largely undetectable in the smallest-order vessels, and is thus more reliably used as a SMC marker. CD146 and RGS5 have very low specificity, and are not common PC markers. The manuscript cited for RGS5 does not demonstrate any immunolabeling of PCs with RGS5. Alkaline phosphatase is described in mural cells undergoing calcification, but is also expressed by SMC and endothelial cells in skeletal muscle. The manuscript cited is controversial and unconvincing, and highlights a general confusion in the field regarding SMCs versus PCs. Nestin is not a PC marker, but is rather an MSC marker. PC stemness and MSC potential is controversial, and has been purported as an artifact of ex vivo culturing (see remark # 3)
7. Section 3.1 presents the contractility of PCs as definitive. In fact, this remains debated in the field. Of the three references listed in Line 127 – 132, two are reviews. The one original research paper cited - Hill et al, 2014 – is a seminal publication, and gave rise to the concept of PC contractility. That same year, another important research manuscript – Hall et al., 2014 – reported that the regional cerebral blood flow was due to SMC contractility, not PCs. It is important to thoroughly present the entire state of the field in a review, which the current authors have not adequately achieved.
8. Section 3.3 accepts the controversial PC attribute of multipotency, without presenting studies that stand in contradiction. Again, out of the 13 references given in this section, only 5 are original research publications.
Overall, although the review is well organized, and includes interesting information regarding signaling, the impact of the review is lacking due to that fact that the authors do not present an impartial, in-depth overview of the topics chosen for focus in this review. The review would greatly benefit form a more in-depth study of the field, and demonstrated effort toward original research citations support.
Author Response
In the review manuscript submitted by Fazio et. al, the authors present an overview of pericyte (PC) identity (including markers), PC functional roles (including contractility, angiogenesis, and stemness), and signaling pathways (related to blood flow, angiogenesis, and stemness). A thorough review each of these areas of PC biology would be a significant contribution to the field in order to provide clarity to a landscape full of debate and controversy. However, the current review does not, in this reviewers perspective, thoroughly review the topics of focus. Over 1/3 of the references provided were of other reviews. This is unacceptable and reflects a lack of effort and sincerity. While signaling mechanisms were better cited and more thoughtfully approached, some sections of the review presented one-side only of controversial topics, further contributing to confusion in the field. In preparing a review, it is necessary to present a thorough overview of the current state of the field, supported by original research publications. The impact of the current review is lacking due to inadequate and limited literature review and citations. For example:
- Wherever possible, it is critical to cite original research papers rather than reviews.
We really thank reviewer 1 for the critical analysis, which significantly enhances the relevance of our research. We have revised the manuscript extensively, significantly reducing the number of review citations. We have focused on incorporating original research papers wherever possible and leave few reviews, particularly in the introductory sections of each topic.
- Lines 41-42 – stemmness of pericytes is controversial, and is not widely accepted among many pericyte scientists. It is important to present both sides of this topic. For instance, PC stemness has been demonstrated as an artifact of cell-culture (Nuno Guimaraes-Camboa et al., 2017, ElenaCano et al 2017) – an environment in which many cell types drift, despite being fully differentiated. Cited with a review.
We really thank reviewer 1 for the insightful suggestions. In response to this feedback, we have tried to present a more balanced perspective on this controversial topic by including viewpoints from original research studies. We recognize that the stemness of pericytes is a controversial topic, so it is crucial to present a balanced view. For this reason, we incorporated the suggested references in order to propose both sides of this debate (L47-54).
- Lines 76 – The authors describe PC morphology as an “elongated cell body aligned with the direction of blood flow”, cited with a review. However, three recognized morphological subtypes of PCs are described in the literature, based on vessel location.
We have updated the manuscript to include a description of the three recognized morphological subtypes of pericytes, as you suggested (L84-88). We appreciate your valuable input in enhancing the completeness and accuracy of our work.
- Lines 82-84 are almost word for word, including references, taken from a 2021 review by Girolamo et al. (although the authors of the review under submission state (incorrectly) neuroectodermal rather than ectomesenchyme. PCs were reported to be derived via neuroectoderm in a 2002 paper by Korn and Kurz, which when examined, actually describes SMCs, not PCs.
As suggested by Reviewer 1, we have corrected the wrong information and changed the reference papers.
- Line 88-89 – NG2 is not just an identifier of developmental PC. It is a prominent PCs marker throughout all stages.
Taking in consideration your correction, we have underlined that NG2 is a prominent marker expressed in all stages of pericyte development and we have added original papers to support it.
- Lines 85-98, and marker Table. The authors include markers that are not widely used or accepted for PC identification, and thus detract from the value of this section. While NG2 and PDGFRβ are prominent PC markers, PC alpha-SMA expression is variable, and largely undetectable in the smallest-order vessels, and is thus more reliably used as a SMC marker. CD146 and RGS5 have very low specificity, and are not common PC markers. The manuscript cited for RGS5 does not demonstrate any immunolabeling of PCs with RGS5. Alkaline phosphatase is described in mural cells undergoing calcification, but is also expressed by SMC and endothelial cells in skeletal muscle. The manuscript cited is controversial and unconvincing, and highlights a general confusion in the field regarding SMCs versus PCs. Nestin is not a PC marker, but is rather an MSC marker. PC stemness and MSC potential is controversial, and has been purported as an artifact of ex vivo culturing (see remark # 3)
Thank you Reviewer 1 for your insightful comments and suggestions. We acknowledge the challenges in accurately characterizing pericytes due to the lack of specific markers. In response to your feedback, we have made revisions to our manuscript to enhance the accuracy and clarity of the marker selection. To address these issues, we have referred to recent key studies. Specifically, we cited the work of Baek et al. (2022) published in Frontiers in Cardiovascular Medicine, which identified specific pericyte markers using single-cell RNA sequencing analysis across various tissues. Additionally, we referred to the study of West et al. (2021) published in Methods in Molecular Biology, which provides a comprehensive research protocol for the immunohistochemical staining of pericytes in a variety of human tissues. Consequently, we have also updated the table related to the markers.
- Section 3.1 presents the contractility of PCs as definitive. In fact, this remains debated in the field. Of the three references listed in Line 127 – 132, two are reviews. The one original research paper cited - Hill et al, 2014 – is a seminal publication, and gave rise to the concept of PC contractility. That same year, another important research manuscript – Hall et al., 2014 – reported that the regional cerebral blood flow was due to SMC contractility, not PCs. It is important to thoroughly present the entire state of the field in a review, which the current authors have not adequately achieved.
We have included both suggested papers underlying the controversial role of pericytes in capillary contractility to provide a comprehensive perspective on this aspect (L168-175).
- Section 3.3 accepts the controversial PC attribute of multipotency, without presenting studies that stand in contradiction. Again, out of the 13 references given in this section, only 5 are original research publications.
Thank you for your valuable feedback. In response to your comment regarding the controversial attribute of multipotency in pericytes, we have expanded the discussion in Section 3.3 to include additional references that present both supportive and contradictory evidence regarding pericyte multipotency (L256-L265). Specifically, we have now incorporated references to studies that indicate limited differentiation potential in certain tissues and contexts, as well as studies that demonstrate multipotent capabilities under specific conditions. This revised discussion provides a more balanced view and acknowledges the existing controversies in the field. We have further emphasized the importance of considering the influence of in vitro conditions on pericyte plasticity and the need for further comprehensive investigations to clarify these findings.
Overall, although the review is well organized, and includes interesting information regarding signaling, the impact of the review is lacking due to that fact that the authors do not present an impartial, in-depth overview of the topics chosen for focus in this review. The review would greatly benefit form a more in-depth study of the field, and demonstrated effort toward original research citations support.
Moreover, we have reviewed the English as requested, and the work has been approved by the two native-speaking co-authors, James A McCubrey and Foteini Dionysia Koufi.

Reviewer 2 Report
Comments and Suggestions for Authors
Please see the attached file for questions and comments, good luck.

Please see the attached file for questions and comments.
Author Response
We really thank reviewer 2 for the critical analysis, which significantly enhances the relevance of our research. We have made the following revisions:
- L74-75: we have corrected the sentence as “This review will begin with an introduction to the fundamental characteristics of pericytes, followed by an exploration of their specific roles and the complex signaling networks that regulate their activities”.
- L82-84: the pericyte identification information provided in the manuscript is sourced from the following articles: 1) C.Rouget,Note sur le developpement de la tunique contractile des vaisseaux, Compt Rend Acad Sci 59 (1874) 559e562; 2)K. Zimmermann, Der feinere Bau der Blutkapillaren, Z Anat. 68 (1923) 29e109.
- L89-91: we have corrected the sentence as “Notably, the ratio of pericytes to endothelial cells varies significantly, ranging from 1:100 in skeletal muscle to 1:3 in the central nervous system, and 1:1 and in the retina”.
- L92: we have changed “multifaceted roles” in “diverse roles”.
- L95: we added the sentence with the subject "pericytes," clarifying that the identification information is specifically related to them.
- We have revised the pericyte markers based on recent research, excluding alkaline phosphatase due to its association with calcification in mural cells. Moreover, its expression by smooth muscle cells and endothelial cells in skeletal muscle makes it too general for pericyte identification. As a result, we have removed it from the table.
- L137-138: we have corrected the sentence as “Pericytes are essential mural cells that significantly contribute to maintaining the blood vessels integrity and orchestrating tissue regeneration and repair processes.”
- L149: we have decided to exclude information about αSMA, recognizing that not all pericytes demonstrate positive staining for this marker.
- 153-155: we have revised the sentence for improved clarity and precision as “Specifically, in the extensively studied neurovascular unit, the interaction between pericytes, endothelial cells, microglia, neurons, and astrocytes is essential to maintain vascular stability and regulate blood flow”.
- L176-180: We have provided a brief description of the function of both Sunitinib and Thalidomide drugs.
- L183: we have corrected the sentence as “No clear evidence in the literature demonstrates the involvement of pericytes in regulating blood flow in skeletal muscle.”
- 1° Comment for section 3.2 angiogenesis: Thank you reviewer 2 for the clarification since it's important to note that pericytes promote angiogenesis by supporting the formation of new blood vessels to supply nutrients and oxygen to the tumor too. Therefore, they are involved in both physiological and pathological conditions.
- L194: we have changed the word “align with” with “line up”.
- 2° Comment for section 3.2 angiogenesis: Pericytes are known as mural cells located within the walls of capillaries and small blood vessels and they are distinct entities from stem cells. Despite this distinction, pericytes do exhibit certain stem cell-like characteristics, particularly their capacity to undergo differentiation into various cell types under specific environmental cues. Notably, this potential differentiation has only been observed in in vitro However, we have corrected the sentence as “where pericytes co-cultured with endothelial cells and astrocytes effectively reorganizethemselves by contributing to the formation of stable capillary-like structures”.
- 3° Comment for section 3.2 angiogenesis: We have referred to angiogenic process since pericytes are able to respond to signals from endothelial cells and the surrounding microenvironment, migrating towards areas of angiogenic activity. Once recruited, pericytes establish direct physical contacts with endothelial cells, contributing to vessel stabilization and maturation. Through interactions with endothelial cells and secretion of various growth factors and signaling molecules, pericytes actively participate in the modulation of vascular permeability and blood flow regulation. Additionally, pericytes have been shown to exert influence on endothelial cell proliferation, migration, and tube formation, thereby influencing the overall process of angiogenesis.
- 1° Comment for section 3.3 Stem cell potential and tissue regeneration: The classification of pericytes into type 1 and type 2 categories primarily allows for the assessment of their differentiation potential rather than their characterization. While this classification helps distinguish pericyte subpopulations based on their differentiation capacity, it does not inherently imply differences in their overall characterization.
- L351-353: we have underlined the specific cellular type in which Akt1 was silenced.
- L360: We have restructured the sentence for better clarity in “Pericytes produce TGF-β, another crucial regulator that exerts a dual role in angiogenesis..”
- L367-369: We have organized the sentence into two for improved clarity.
- L418-420: We have restructured the sentence by removing “not only”-“but also”.
- Moreover, we have reviewed the English as requested, and the work has been approved by the two native-speaking co-authors, James A McCubrey and Foteini Dionysia Koufi.

Round 2
Reviewer 1 Report
Comments and Suggestions for Authors
The authors made significant progress in addressing concerns. That contributes to the overall impact of the review. However, the authors continue to make declarative and assumptive statements throughout the review. Please soften such language.
1. Section 2 - Alpha-SMA is highly expressed by smooth muscle cells. Please reflect this in lines 117-18, and in Table 1.
2. Section 3.1 - The efforts of the authors to include non-contractile perspective is appreciated. However, pericyte contraction remains presented as fact. Please soften the language in lines 177-179.
3. Section 3.1 – lines 198 – 205. It is good to include this perspective, however, the author describes the discrepancy as possibly due to a poor definition of pericytes. While this is the case in many pericyte-related manuscripts, it is not an accurate characterization of the Hill paper. Please address this. Further, despite including this new data, the author does not review the topic impartially. The Hill paper data description is followed by lines (206-213) that re-emphasize PC regulation of blood flow, supported by data centered around PDGFR interventions/manipulations – a cell surface receptor that is expressed not only by pericytes, but also smooth muscle cells, which do, in fact, regulate blood flow. Nevertheless, the impacts of these PDGFR-based interventions on blood-flow (on very large vessels) are attributed to pericytes by the authors (not smooth muscle cells, or even general mural cells). This is misleading.
4. In general, surrounding controversial topics, please soften the language. For instance, instead of stating that “pericytes can differentiate into other cell lineages” (line 284), soften to “ It is reported that pericytes may differentiate into…”. Absolute statements, and statements that make assumptions (ie. Line 480 - “Leveraging their stem cell-like properties”, rather than “….their potential stem cell-like properties…”) are made throughout the manuscript and contribute to misconception and misinformation within the field. Although it is easy to present data as black and white, it rarely is, and it is critical to use language that reflects the uncertainties inherent within the scientific process. A good review should thoroughly and accurately present the current state of the literature in an unbiased manner, allowing the reader to consider both sides of existing controversies, rather than be influenced (or mislead) toward an agenda. Please address the absolute statements made throughout the review.
Author Response
The authors made significant progress in addressing concerns. That contributes to the overall impact of the review. However, the authors continue to make declarative and assumptive statements throughout the review. Please soften such language.
- Section 2 - Alpha-SMA is highly expressed by smooth muscle cells. Please reflect this in lines 117-18, and in Table 1.
As suggested by reviewer 1, we have updated lines 117-118 and Table 1 to reflect that α-SMA is also expressed in these cells.
- Section 3.1 - The efforts of the authors to include non-contractile perspective is appreciated. However, pericyte contraction remains presented as fact. Please soften the language in lines 177-179.
Thank you for your valuable feedback. As suggested, we have softened the language in lines 177-179 (in the new draft are L146-148) focusing on the general role of pericytes in microvascular blood flow.
- Section 3.1 – lines 198 – 205. It is good to include this perspective, however, the author describes the discrepancy as possibly due to a poor definition of pericytes. While this is the case in many pericyte-related manuscripts, it is not an accurate characterization of the Hill paper. Please address this. Further, despite including this new data, the author does not review the topic impartially. The Hill paper data description is followed by lines (206-213) that re-emphasize PC regulation of blood flow, supported by data centered around PDGFR interventions/manipulations – a cell surface receptor that is expressed not only by pericytes, but also smooth muscle cells, which do, in fact, regulate blood flow. Nevertheless, the impacts of these PDGFR-based interventions on blood-flow (on very large vessels) are attributed to pericytes by the authors (not smooth muscle cells, or even general mural cells). This is misleading.
Following the suggestion from reviewer 1, we have revised the paragraph spanning from L165-177 to present two contrasting perspectives while emphasizing the necessity for further studies, including investigations into surrounding cell types. We have also removed the inaccurate statement regarding the "poor definition of pericytes”.
- In general, surrounding controversial topics, please soften the language. For instance, instead of stating that “pericytes candifferentiate into other cell lineages” (line 284), soften to “ It is reported that pericytes may differentiate into…”. Absolute statements, and statements that make assumptions (ie. Line 480 - “Leveraging their stem cell-like properties”, rather than “….their potential stem cell-like properties…”) are made throughout the manuscript and contribute to misconception and misinformation within the field. Although it is easy to present data as black and white, it rarely is, and it is critical to use language that reflects the uncertainties inherent within the scientific process. A good review should thoroughly and accurately present the current state of the literature in an unbiased manner, allowing the reader to consider both sides of existing controversies, rather than be influenced (or mislead) toward an agenda. Please address the absolute statements made throughout the review.
We have reviewed the text and made the necessary adjustments within the manuscript to soften the language surrounding controversial topics.